# Improving the Structural Characteristics of Heavy Concrete by Combined Disperse Reinforcement

**Levon R. Mailyan** [1], **Alexey N. Beskopylny** [2,*], **Besarion Meskhi** [3], **Aleksandr V. Shilov** [4], **Sergey A. Stel'makh** [5], **Evgenii M. Shcherban'** [5], **Alla S. Smolyanichenko** [6] and **Diana El'shaeva** [7]

[1] Department of Roads, Don State Technical University, 344000 Rostov-on-Don, Russia; lrm@aaanet.ru
[2] Department of Transport Systems, Faculty of Roads and Transport Systems, Don State Technical University, Gagarin, 1, 344000 Rostov-on-Don, Russia
[3] Department of Life Safety and Environmental Protection, Faculty of Life Safety and Environmental Engineering, Don State Technical University, Gagarin, 1, 344000 Rostov-on-Don, Russia; reception@donstu.ru
[4] Department of Reinforced Concrete Structures, Faculty of Industrial and Civil Engineering, Don State Technical University, Gagarin, 1, 344000 Rostov-on-Don, Russia; avshilov75@mail.ru
[5] Department of Engineering Geology, Bases, and Foundations, Don State Technical University, 344000 Rostov-on-Don, Russia; sergej.stelmax@mail.ru (S.A.S.); au-geen@mail.ru (E.M.S.)
[6] Department of Water Supply and Sewerage, Don State Technical University, 344000 Rostov-on-Don, Russia; arpis-2006@mail.ru
[7] Department of Technological Engineering and Expertise in the Construction Industry, Don State Technical University, 344000 Rostov-on-Don, Russia; diana.elshaeva@yandex.ru
* Correspondence: besk-an@yandex.ru; Tel.: +7-8632738454

**Abstract:** The development of perspective concrete mixes capable of resisting the action of external loads is an important scientific problem in the modern construction industry. This article presents a study of the influence of steel, basalt, and polypropylene fiber materials on concrete's strength and deformation characteristics. A combination of various types of dispersed reinforcement is considered, and by methods of mathematical planning of the experiment, regression dependences of the strength and deformation characteristics on the combination of fibers and their volume fraction are obtained. It was shown that the increase in compressive strength was 35% in fiber-reinforced concretes made using a combination of steel and basalt fiber with a volume concentration of steel fiber of 2% and basalt fiber of 2%; tensile strength in bending increased by 79%, ultimate deformations during axial compression decreased by 52%, ultimate deformation under axial tension decreased by 39%, and elastic modulus increased by 33%. Similar results were obtained for other combinations of dispersed reinforcement. The studies carried out made it possible to determine the most effective combinations of fibers of various types of fibers with each other and their optimal volume concentration.

**Keywords:** fiber-reinforced concrete; hybrid reinforcement; compressive strength; tensile strength; ultimate deformations; optimal composition

## 1. Introduction

At present, dispersion-reinforced concrete is undergoing significant development in the construction field. This is related to the fact that, with all the benefits, concrete and reinforced concrete has several disadvantages. The most severe drawback is considered to be low crack resistance, which is the cause of the brittle fracture of structures. The brittle fracture of concrete is the most dangerous because it can lead to the sudden and progressive destruction of an entire building or structure. The research for ways to eliminate this and other shortcomings is an important scientific problem, and one of the ways to solve this task is to use fiber-reinforced concrete—a composite material consisting of a cement matrix with a uniform or specified distribution over the entire volume of oriented or randomly located discrete fibers of different sizes.

Polymers have proven to be excellent fiber fillers. In [1], the advantages of using reinforcing fibers in latex-modified, fast-set cement concrete for emergency road structural layers repair were considered and studied in terms of strength, permeability and durability depending on the type of fiber. Uniform fibers were evaluated, including jute, PVA and nylon fibers, as well as hybrid blends of these PVA, and nylon fibers in a 1:1 weight ratio. The road surface in need of repair was dismantled and replaced with coarse aggregate. The fast-setting binder, fibers, and latex were then mixed and placed on a coarse aggregate layer. Hybrid fiber-reinforced concrete made of PVA and nylon fibers showed the best properties when used as a road surface repair composition [1].

High-Performance Fiber-Reinforced Concrete (HPFRC) technologies are discussed in Review [2]. The addition of short discrete fibers into concrete can be used to resist and prevent crack propagation—the effect of fibers on the properties of concrete. This article deals with the problems of the mechanism of formation and propagation of cracks, the behavior of the stress–strain state, tensile strength (TS) and other properties of HPFRC. In general, it has been shown that the addition of fibers to high-quality concrete improves the mechanical properties of concrete, especially TS, flexural strength and ductility.

Evaluation of the ultimate strength in the bending of concrete elements reinforced with ultra-high-performance fiber (UHPFRC) is carried out in [3]. Experimental studies included testing beams, taking into account the effect of the fiber volume fraction, the shear length to depth ratio and the compressive strength of the matrix. The authors found that the inclusion of steel fiber with a volume fraction of 2% significantly increases the shear and flexural strength of the ultra-high foam fiber beams. The addition of steel fibers changed the failure behavior of the beams from diagonal shear failure to flexural failure. To accurately predict the shear strength of UHCF beams, the tensile strength of the matrix and the effect of steel fibers must be considered.

In recent years, the direction of geopolymer fiber-reinforced concrete [4–6] has been actively developing. In [7], the results of studies of the ultimate axial load and bending moment of simple and fiber-reinforced geopolymer columns (GPC, FRGPC) are presented. Fiber-reinforced concrete columns with steel and synthetic fibers were reinforced with double layers of longitudinal and transverse reinforcement using steel and/or glass fiber-reinforced polymer (GFRP) rods. The performance of alkali-activated cinder concrete (AAC) reinforced with structural polypropylene and steel fibers cured at room temperature was studied in [8]. Structural polypropylene and steel fibers were included in the alkali-activated mix at 1.5% and 5% of the total binder weight, respectively. Breaking strength increased significantly by 19.28% and 26.80% due to the inclusion of structural polypropylene and steel fibers, respectively.

The study of the influence of the morphology of the coarse aggregate on the properties of self-compacting high-performance fiber-reinforced concrete (SCHPFRC) was carried out in [9]. The results obtained showed that the morphology of the coarse aggregate grains significantly affects the strength, deformation, and rheological properties of SCHPFRC.

In [10], an experimental analysis of the bending behavior of reinforced concrete beams reinforced with fibrous polymer materials using surface mounting and external bonding methods is presented. Six double-span beams with a total length of 3200 mm and a 120/200 mm cross-section were tested under short-term and monotonically increasing loads. The authors obtained the results of an increase in the maximum carrying capacity of reinforced beams by 22–82% compared to an unreinforced control beam. A significant result is information that the ductility of the beams reinforced with CFRP rods was satisfactory, while the ductility of the beams reinforced with CFRP rods was very low; therefore, the nature of the destruction of these beams was fragile.

The assortment of fibers used is very extensive, and, according to the accepted classification, they are divided:

- By the modulus of elasticity of the fiber—into high-modulus (steel, carbon, glass, etc.) and low-modulus (polypropylene, viscose, etc.);
- By origin—into natural (asbestos, basalt, wool, etc.) and artificial (viscose, polyamide, etc.);

- For the main material—into metal (most often steel) and non-metal (synthetic, mineral).

In dispersed reinforcement, concrete hardening with fiber is based on the hypothesis that the composite matrix transfers the applied load to the fibers uniformly distributed in it due to the tangential forces that act on the interface. If the fiber elastic modulus exceeds the modulus of elasticity of the concrete matrix, the main part of the stresses is absorbed by the fibers, and the total strength of the composite is directly proportional to the fiber volume fraction.

It is known that the main feature of composites, including fiber-reinforced concretes, is the heterogeneity, which determines the complexity of the structure of such materials [11,12].

Fiber-reinforced concrete is a structure-in-structure type material with a very complex poly-structural organization, in which at least two scale levels can be distinguished:

- Microscopic (the level of cement stone), which establishes the phase composition of the neoplasms, the type, nature of porosity, etc.
- Macroscopic (concrete level), which establishes the type and properties of aggregate, cement stone, fiber and the ratio between them, and the uniformity of distribution of these components in the volume of fiber-reinforced concrete.

The structural-technological model of fiber-reinforced concrete was proposed in [13], a component of which is a macrostructural cell, the dimensions of which depend on the degree of saturation with the reinforcing fiber and are commensurate with the geometric characteristics of the fibers and the dimensions of the filler. All components that make the macrostructural cell are interconnected by contacts, the strength of which determines the basic properties of dispersion-reinforced concrete.

Depending on the place of formation, contacts can be divided into three types: between cement grains; between cement stone and aggregate; between fine-grained concrete and fiber. Cement stone, aggregate and fiber do not occupy the entire volume, since various pores remain, which may contain capillary and free water, as well as air [14–16].

A distinctive feature of dispersed-reinforced concretes is the presence of discrete fibers in their composition, the effect of which on the changes occurring in the structure and properties of the material must be taken into account when assigning the composition of the concrete matrix [17–20].

The fundamental principle used in the development of private techniques is the concept of fiber as a part of a kind of filler with a developed surface and has a significant impact on the technological characteristics of the mixture of components [21–24].

Many scientists dealing with dispersed-reinforced concrete agree with the hypothesis that the critical factor that determines the properties of fiber-reinforced concrete is the adhesion strength of dispersed reinforcement to the matrix. However, the characteristics by which the adhesion strength can be judged, as well as the methods for their determination, are offered differently. Thus, in [25], a coefficient of utilization of the strength characteristics of reinforcing fibers $K_{isp}$ is proposed, which varies from 0 to 1

$$K_{isp} = \frac{R_{drc} - R_M}{k_x \cdot \mu_0 \cdot R_a} \tag{1}$$

where $R_{drc}$ is the ultimate strength of dispersed-reinforced concrete; $R_m$ is the ultimate strength of the matrix; $k_x$ is the fiber orientation coefficient; $\mu_0$—the degree of saturation of the concrete matrix with fibers; $R_a$—ultimate strength of reinforcement.

In [13], a method was developed for the characteristic of fibers' adhesion with a cement stone from a normal mortar concentration

$$\tau = \frac{R_{fc} - 3,5 * R_{cz} * \mu_{min} - (1 - 4,5 * \mu_{min}) * R_{cc}}{2 * \frac{l}{d} * \mu_{min}} \tag{2}$$

where $R_{fc}$ is the strength of fiber cement; $R_{cz}$—the strength of the contact zone; $R_{cc}$—the strength of cement stone from a normal mortar concentration; $\mu_{min}$ is the minimum reinforcement coefficient.

Thus, the problem of designing compositions of fiber-reinforced concrete is reduced to the rational choice of dispersed reinforcement, corresponding to the conditions of its operation and purpose. This means that when formulating a task for the design of fiber-reinforced concrete compositions, it is necessary to specify the following information:

- The size and type of the product;
- Ultimate tensile strength in bending, ultimate strength in compression, crack resistance, and fracture toughness;
- Concrete workability (rigidity or mobility);
- Frost resistance, abrasion, water resistance, and other characteristics [26–29].

It is known that dispersed concrete reinforcement with high modulus fibers (steel, carbon, etc.) increases the concrete strength. To the greatest extent, dispersed reinforcement with high-modulus fiber makes it possible to increase the tensile strength of fiber-reinforced concrete in bending. According to known data [30–33], the tensile strength of concrete with steel fiber reinforcement, the diameter of 0.3 mm, regarding 3% fiber reinforcement by volume, increases five times compared to unreinforced concrete. In [34,35], the volume fiber reinforcement of concrete is 2%, and the flexural strength increases two times. Using a steel fiber with a larger diameter, it is also possible to achieve higher tensile strength several times. Evaluation of the diffusion coefficient of chloride ions in self-compacting concrete with steel fibers was carried out in [36]. The article presents three procedures for calculating the diffusion coefficient—rapid chloride penetration test, accelerated chloride penetration test, and measurement of surface electrical resistivity using a Wenner probe.

The addition of synthetic fibers to the concrete mixture does not lead to a noticeable increase in the strength of the composite for axial tension, elongation in bending, and compression under the action of static loads, since concrete cannot transfer static forces to fibers, which have a lower modulus of elasticity compared to concrete. However, despite the low values of elastic characteristics, in comparison with steel, polypropylene fibers are still of great interest in terms of their use for dispersed reinforcement [37–42].

In [43], hybrid fibers such as steel and polypropylene were added to the cement with a high fly-ash content to improve mechanical properties. The authors combined silane-coated basalt fiber with steel and polypropylene due to its availability and low cost. Various combinations of hybrid fibers (both two and three types) were tested on mortar samples to determine the optimum fiber percentage. It was found that the compressive strength of the three types of hybrid fiber-reinforced concrete increased markedly by 5.44%, and the increase in tensile strength when splitting was 6.77% compared to the hybrid of the two types.

As a rule, with dispersed reinforcement, mono-reinforcement is the main option, in which the control of concrete properties is rather limited. Then, the combined reinforcement (reinforcement with different types of fibers) makes it possible to control a wide range of properties in one composite. At the same time, the issues of the combined reinforcement of fiber-reinforced concrete have not been studied enough to date, and the available information testifies to the inconsistency of the research results obtained, which reduces the volume of use of dispersed reinforcement.

Thus, the main aim of this article is to study the physical-mechanical, energy and deformative characteristics and develop promising options for the dispersed combined reinforcement of fiber-reinforced concrete to increase the technical and economic efficiency, reliability and operational safety of buildings and structures.

## 2. Materials and Methods

To study the impact of a combination of different types of fibers on the strength and deformation characteristics of fiber-reinforced concrete, a program of experimental studies was developed. The block diagram of the experiment program is shown in Figure 1.

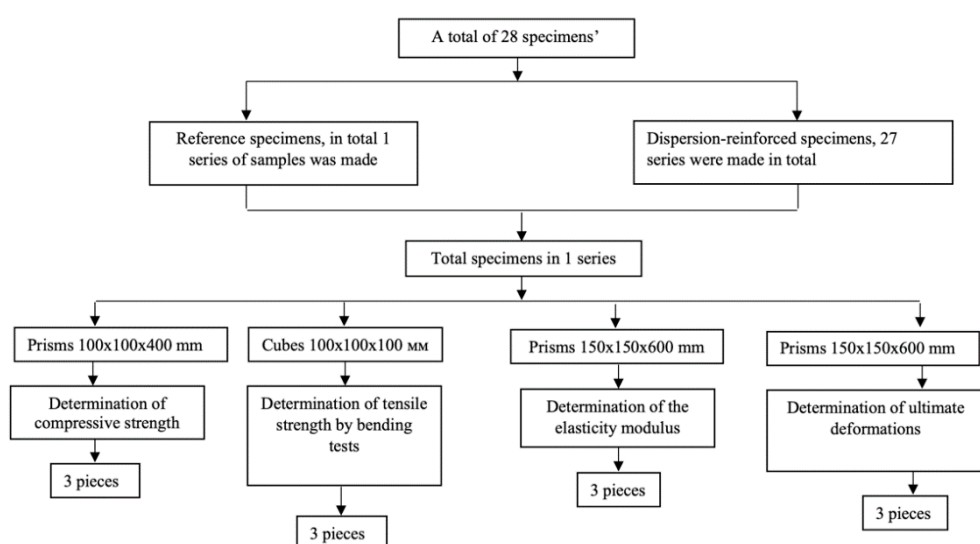

**Figure 1.** Block diagram of the experiment program.

During the research, the no-additive Portland cement of the PC500 D0 brand produced by OAO "Novoroscement" was used. Table 1 shows Portland cement's physical and mechanical characteristics, and Table 2 shows the chemical and mineralogical composition of Portland cement.

**Table 1.** Characteristics of Portland cement PC500 D0.

| Indicator Title | Value |
|---|---|
| Specific surface area (according to Blaine) | $\geq$340 m$^2$/kg |
| Oversize product # 008 | $\leq$3.5% |
| Mortar standard consistency, % | $\leq$26.5 |
| Setting time, hour-min: | |
| - start | not earlier 2–00 |
| - finish | not later 6–00 |
| Tensile strength during the bending test, MPa: | |
| - 2 days | $\geq$4.0 |
| - 28 days | $\geq$6.0 |
| Compressive strength, MPa: | |
| - 2 days | $\geq$21.0 |
| - 28 days | $\geq$52.0 |

**Table 2.** Mineralogical and chemical composition of Portland cement 500 D0.

| Grade of Cement | Mineralogical Composition, No More % | | | Chemical Composition | | |
|---|---|---|---|---|---|---|
| | $C_3S$ | $C_3A$ | $C_3A + C_3AF$ | Chlorinity Content CL, in Cement | Mass Fraction of Alkaline Oxides $R_2O$, in Clinker | Content of Magnesium Oxide, MgO in Clinker |
| 500 D0 | 60 | 5.0 | 20% | $\leq$0.02% | $\leq$0.9% | $\leq$0.9% |

River sand was used as a fine aggregate in accordance with GOST 8736 "Sand for construction works. Specifications" [44]. The granulometric composition of the sand is presented in Table 3, and the main physical and mechanical characteristics are presented in Table 4.

**Table 3.** Granulometric composition of sand.

| Scheme | Sieve Diameter, mm | | | | | | Fineness Modulus |
|---|---|---|---|---|---|---|---|
| | 2.5 | 1.25 | 0.63 | 0.315 | 0.16 | <0.16 | |
| partial | 1.89 | 4.71 | 24.52 | 59.18 | 6.67 | 0.31 | 2.27 |
| full | 1.89 | 6.60 | 31.13 | 90.30 | 96.97 | | |

**Table 4.** Physical and mechanical properties of sand.

| Content of Dust and Clay Particles, % | Clay Content in Lumps, % | Bulk Density, kg/m$^3$ | True Density, kg/m$^3$ | The Organic Inclusions Presence |
|---|---|---|---|---|
| 0.7 | 0.16 | 1632 | 2650 | absent |

Steel, basalt, and polypropylene fibers were used as dispersed reinforcement. Table 5 shows the physical and mechanical characteristics of the fibers used. The fibers are shown in Figures 2–4.

**Table 5.** Physical and mechanical properties of fiber (data were received from manufacturer).

| Fiber Type | Tensile Strength, MPa | Fiber Diameter, m | Fiber Length, mm | Elastic Modulus GPa | Density, g/cm$^3$ | Elongation Ratio, % |
|---|---|---|---|---|---|---|
| Basalt | 3500 | $16 \times 10^{-6}$ | 12 | 3500 | 2.6 | 3.2 |
| Polypropylene | 500 | $15 \times 10^{-6}$ | 20 | 35 | 0.91 | 120 |
| Steel | 1500 | $0.3 \times 10^{-3}$ | 30 | 190 | 7.8 | 3 |

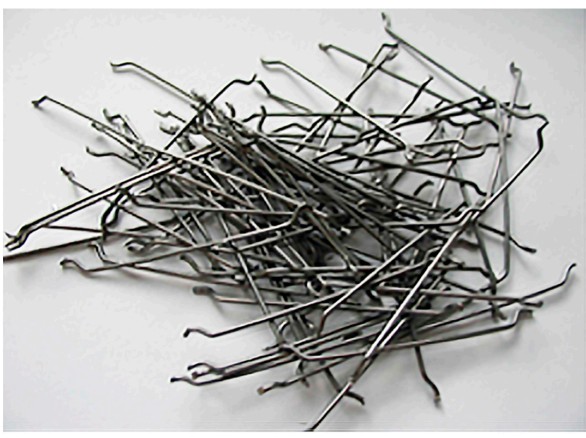

**Figure 2.** Steel fiber (scale 1:1).

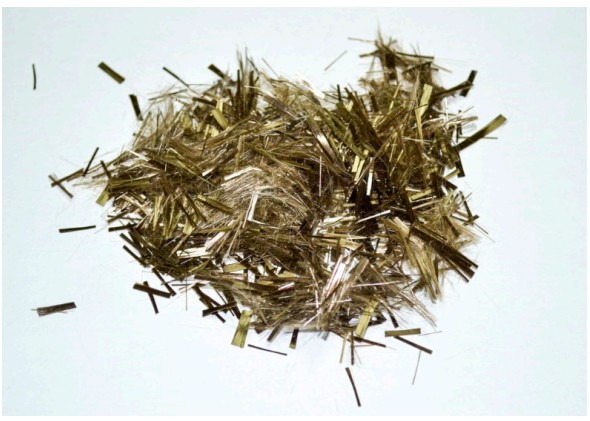

**Figure 3.** Basalt fiber (scale 1:1).

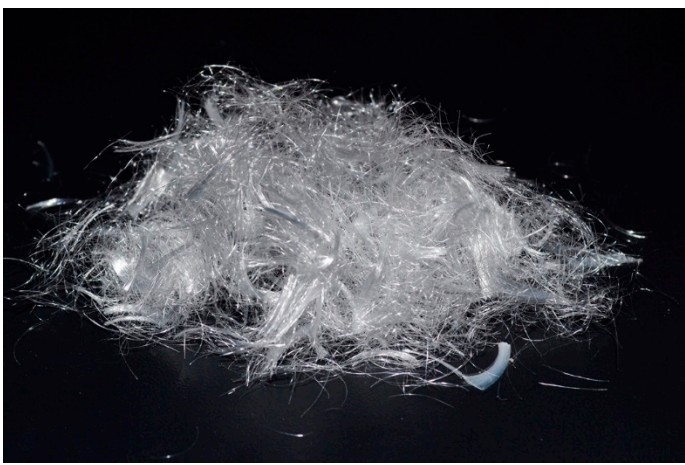

**Figure 4.** Polypropylene fiber (scale 1:1).

The superplasticizer Schomburg Remicrete SP-10 (FM) was used to regulate the mobility of concrete mixtures. It is a highly effective polyester carboxylate-based plasticizer that contributes to accelerated early and final strength development by acting on the hydration processes. The technical characteristics of the superplasticizer are shown in Table 6.

**Table 6.** Superplasticizer specifications.

| Superplasticizer Schomburg Remicrete | Raw Material Base | Colour | The Physical State | Density, g/cm³ |
|---|---|---|---|---|
| SP-10 (FM) | polyether carboxylate | light yellow | liquid | 1.1 |

All samples were made of fine-grained concrete of the same composition Cement/Sand = 1:2 at Water/Cement = 0.32 and the consumption of the superplasticizer additive in the amount of 0.6% of the cement mass.

A laboratory concrete mixer BL-10 was used (Figure 5).

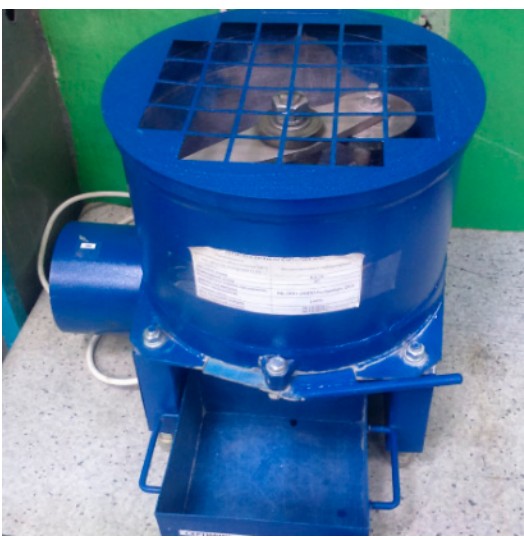

**Figure 5.** Laboratory concrete mixer BL-10.

The mixing of the components of the concrete mixture was carried out in the following sequence. At first, sand and Portland cement were loaded into the laboratory concrete mixer.

Then, water with an additive, which was previously dissolved in it, was introduced. When the cement–sand mortar was ready, fiber was added, and the mixing process continued until the uniform distribution of fiber fibers throughout the volume was ensured. For the manufacture of cubes, standard forms of 2FK-100 grade were used, for prism samples, forms of the FP-100 and FP-150 grades. Compaction of the fiber-reinforced concrete mixture in the process of forming the samples was carried out on a laboratory vibrating platform SMZH—539—220 A with mechanical fastening. The vibration time averaged 60–90 s. On the next day after molding, the samples were demolded and placed in a normal hardening chamber for 28 days until the design strength was achieved [45–49].

For research, testing equipment (hydraulic press IPS-10, URI device for tensile testing in the bending of concrete beams) and measuring instruments (measuring metal ruler, laboratory scales, device for measuring deviations from the plane NPL-1, device for measuring deviations from the perpendicularity of the NPR-1, a chain of strain gauges with a base of 50 mm and dial indicators with the graduation of 0.001 mm) were also used.

Compression and tensile bending tests were carried out in accordance with the requirements of Russian standard GOST 10180 "Concretes. Methods for strength determination using reference specimens" [50] (European standard EN 12390 "Testing hardened concrete").

Axial compression and axial tension tests were carried out in accordance with the requirements of Russian standard GOST 24452 "Concretes. Methods of prismatic, compressive strength, modulus of elasticity and Poisson's ratio determination" [51].

Devices for measuring deformations of samples should be installed along four of its edges or along three or four generatrices of the cylinder, turned at an angle of 120° or 90°. Devices for measuring transverse deformations should be installed in the middle of the sample height normal to the bases of measuring longitudinal deformations.

For fixing the indicators, devices are used in the form of steel frames, fixed to the sample with four stop screws—two on opposite sides of the sample—or supporting inserts glued to the sample.

Frames should be made of steel strips, support inserts—from steel squares or rods with holes for fixing indicators. As a connecting insert for measuring longitudinal deformations, connecting inserts–frames should be used, which provide the ability to measure deformations until the end of the destruction of the sample.

A fast-curing, low-swelling adhesive should be used to secure the support inserts.

Before gluing, the surface of the sample should be degreased with an organic solvent, and then the support insert should be heated to a temperature of 50–60 °C. The support insert in a hot state is pressed against the surface of the sample, having previously applied glue to it.

Additionally, calculations of the strength characteristics and the modulus of elasticity of fiber-reinforced concrete were carried out depending on the combination of various types of fibers and their volume content. The calculations were carried out by the method of mathematical planning of the experiment (PFE 2k) using the MathCAD program.

## 3. Results and Discussion

### 3.1. Results of Experimental Studies of the Influence of a Combination of Various Types of Fibers and Their Volumetric Content on the Strength and Deformation Characteristics of Fiber-Reinforced Concrete

Determination of the strength and deformation characteristics of fiber-reinforced concrete was carried out using regression dependences, the type and values of the coefficients of which are determined by the methods of mathematical experimental design techniques.

The strength and deformation characteristics of fiber-reinforced concrete are taken as the response functions, depending on the combination of different types of fibers $k$ and their volume content $\mu$.

The response functions:

- $R_c$ $(k, \mu)$—compressive strength, MPa;
- $R_{tben}$ $(k, \mu)$—tensile strength in bending test, MPa;

- $\varepsilon_c\ (k,\ \mu)$—ultimate deformations during axial compression, mm/m * $10^{-3}$;
- $\varepsilon_{btR}\ (k,\ \mu)$—ultimate deformations during axial tension, mm/m * $10^{-4}$;
- $E_{fb}\ (k,\ \mu)$—elastic modulus, MPa * $10^3$.

As arguments, the types of fiber combinations *k* and their volume content μ were taken in absolute terms with different levels of variation. The values of the variables are presented in Table 7.

**Table 7.** Values varying factors.

| N. | Factor | Levels | | |
| --- | --- | --- | --- | --- |
| | | **Minimal** | **Zero** | **Maximum** |
| 1 | Fiber combination type (k) | steel/basalt 0/0 | propylene/basalt 0/0 | steel/propylen 0/0 |
| 2 | Fiber volume content (μ), % | 0.5/1.5 1.0/1.0 1.5/0.5 | 0.5/2.5 1.5/1.5 2.5/0.5 | 1.5/2.5 2.0/2.0 2.5/1.5 |

A photo illustration of the compressive strength test of cubes of basic composition is shown in Figure 6.

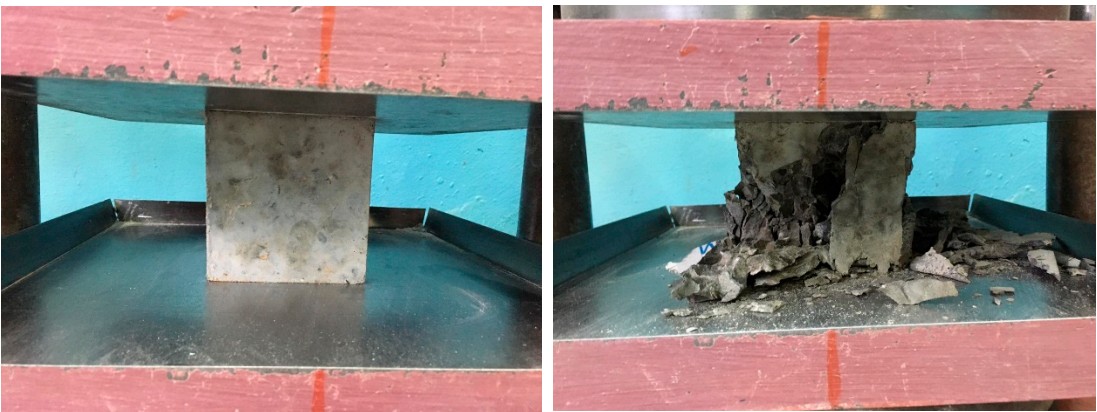

**Figure 6.** Compressive strength test of cubes of basic composition.

The results of experimental studies of the effect of a combination of various types of fibers and volume content on the strength and deformation characteristics of fiber-reinforced concrete are presented in Table 8 and Figures 7–11.

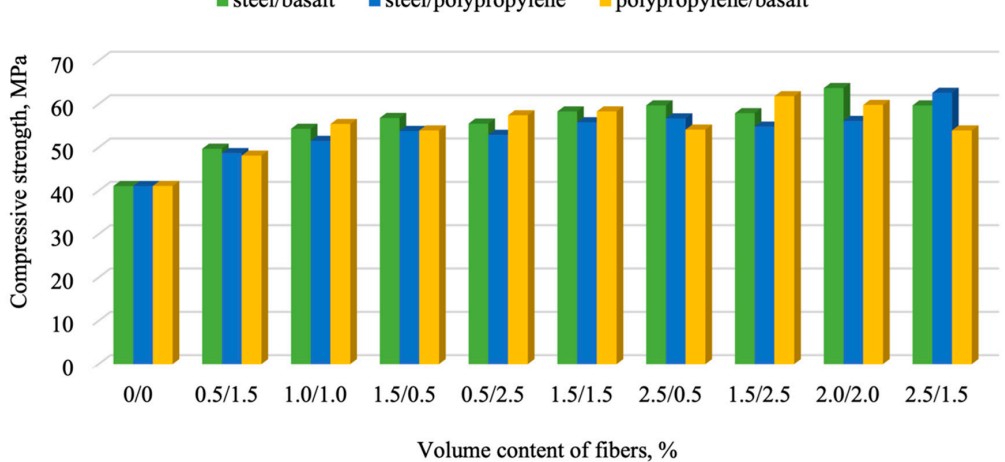

**Figure 7.** Dependence of the compressive strength of fiber-reinforced concrete on the type of fiber combination and volume content.

**Table 8.** Results of experimental studies of the effect of a combination of various types of fibers and volume content on the strength and deformation characteristics of fiber-reinforced concrete.

| N | Fiber Type Combination | Fiber Volume Content (μ), % | Compressive Strength, MPa | Tensile Strength in Bending Test, MPa | Ultimate Deformations during Axial Compression, mm/m*$10^{-3}$ | Ultimate Deformations during Axial Tension, mm/m *$10^{-4}$ | Elastic Modulus, GPa |
|---|---|---|---|---|---|---|---|
| 1 | steel/basalt | 0/0 | 41.1 | 4.3 | 2.11 | 1.33 | 32.9 |
| | | 0.5/1.5 | 49.7 | 14.4 | 1.42 | 0.78 | 39.2 |
| | | 1.0/1.0 | 54.3 | 14.8 | 1.51 | 0.73 | 39.8 |
| | | 1.5/0.5 | 56.8 | 15.1 | 1.36 | 0.71 | 40.8 |
| 2 | steel/polypropylene | 0/0 | 41.1 | 4.3 | 2.11 | 1.33 | 32.9 |
| | | 0.5/1.5 | 48.7 | 12.5 | 1.56 | 0.85 | 38.7 |
| | | 1.0/1.0 | 51.5 | 14.8 | 1.48 | 0.72 | 40.1 |
| | | 1.5/0.5 | 53.8 | 15.9 | 1.33 | 0.70 | 41.3 |
| 3 | steel/basalt | 0/0 | 41.1 | 4.3 | 2.11 | 1.33 | 32.9 |
| | | 1.5/2.5 | 57.9 | 17.1 | 1.23 | 0.63 | 44.9 |
| | | 2.0/2.0 | 63.7 | 21.2 | 1.11 | 0.52 | 49.3 |
| | | 2.5/1.5 | 59.7 | 18.3 | 1.21 | 0.57 | 46.5 |
| 4 | steel/polypropylene | 0/0 | 41.1 | 4.3 | 2.11 | 1.33 | 32.9 |
| | | 1.5/2.5 | 54.8 | 16.9 | 1.31 | 0.69 | 44.5 |
| | | 2.0/2.0 | 56.1 | 17.3 | 1.24 | 0.61 | 45.1 |
| | | 2.5/1.5 | 62.6 | 19.3 | 1.15 | 0.55 | 48.2 |
| 5 | steel/basalt | 0/0 | 41.1 | 4.3 | 2.11 | 1.33 | 32.9 |
| | | 0.5/2.5 | 55.5 | 14.8 | 1.52 | 0.73 | 46.9 |
| | | 1.5/1.5 | 58.3 | 17.9 | 1.25 | 0.60 | 45.2 |
| | | 2.5/0.5 | 59.7 | 18.3 | 1.22 | 0.58 | 46.1 |
| 6 | steel/polypropylene | 0/0 | 41.1 | 4.3 | 2.11 | 1.33 | 32.9 |
| | | 0.5/2.5 | 52.9 | 13.9 | 1.47 | 0.80 | 38.9 |
| | | 1.5/1.5 | 55.8 | 16.4 | 1.34 | 0.70 | 44.3 |
| | | 2.5/0.5 | 56.7 | 17.9 | 1.27 | 0.59 | 45.4 |
| 7 | polypropylene/basalt | 0/0 | 41.1 | 4.3 | 2.11 | 1.33 | 32.9 |
| | | 0.5/1.5 | 48.1 | 15.1 | 1.34 | 0.72 | 40.9 |
| | | 1.0/1.0 | 55.4 | 16.8 | 1.29 | 0.69 | 44.7 |
| | | 1.5/0.5 | 53.9 | 14.9 | 1.45 | 0.75 | 40.1 |
| 8 | polypropylene/basalt | 0/0 | 41.1 | 4.3 | 2.11 | 1.33 | 32.9 |
| | | 1.5/2.5 | 61.8 | 18.9 | 1.19 | 0.56 | 47.3 |
| | | 2.0/2.0 | 59.8 | 17.1 | 1.22 | 0.59 | 44.9 |
| | | 2.5/1.5 | 53.9 | 16.1 | 1.31 | 0.70 | 44.1 |
| 9 | polypropylene/basalt | 0/0 | 41.1 | 4.3 | 2.11 | 1.33 | 32.9 |
| | | 0.5/2.5 | 57.4 | 17.7 | 1.26 | 0.62 | 45.6 |
| | | 1.5/1.5 | 58.3 | 18.1 | 1.19 | 0.59 | 46.0 |
| | | 2.5/0.5 | 54.1 | 16.8 | 1.28 | 0.68 | 44.1 |

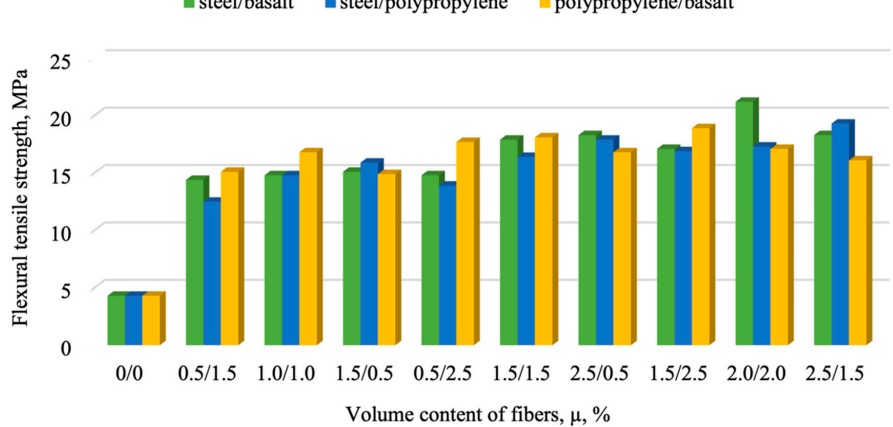

**Figure 8.** Dependence of the tensile strength in the bending of fiber-reinforced concrete on the type of fiber combination and volume content.

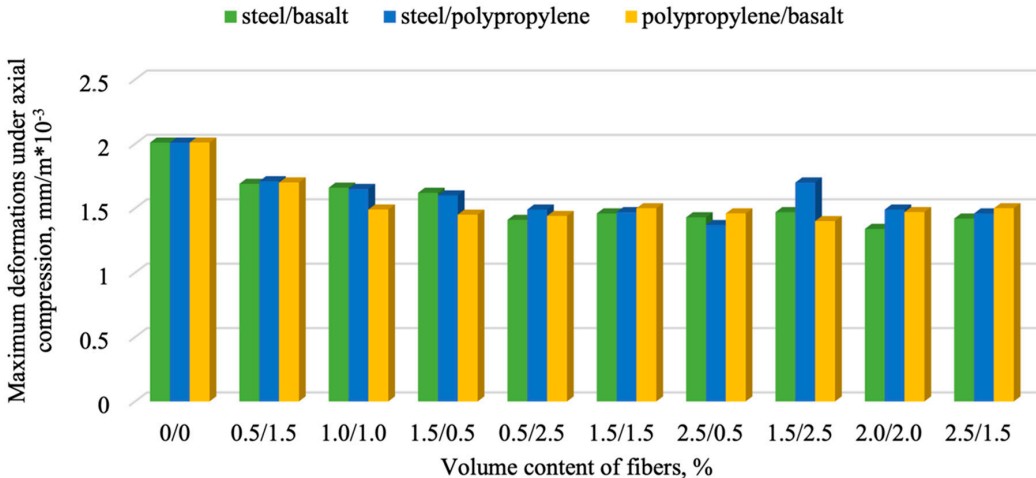

**Figure 9.** Dependence of the ultimate deformations of fiber-reinforced concrete under axial compression on the type of fiber combination and volume content.

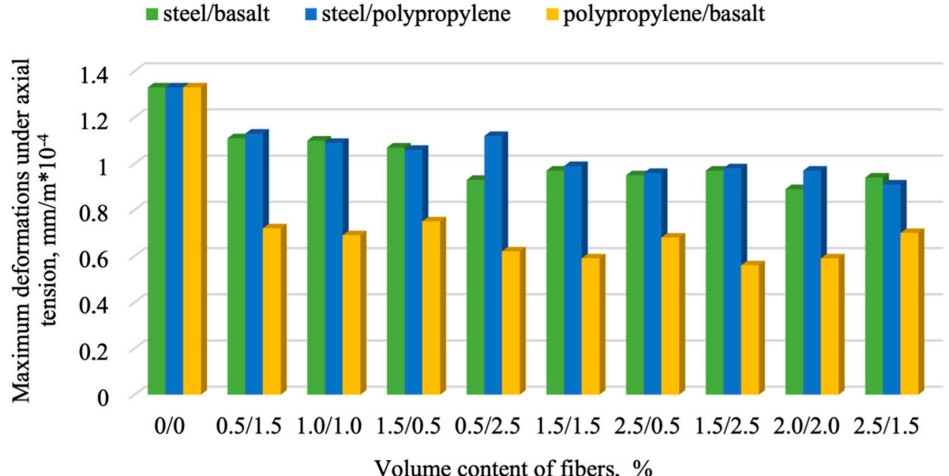

**Figure 10.** Dependence of ultimate deformations of fiber-reinforced concrete under axial tension on the type of fiber combination and volume content.

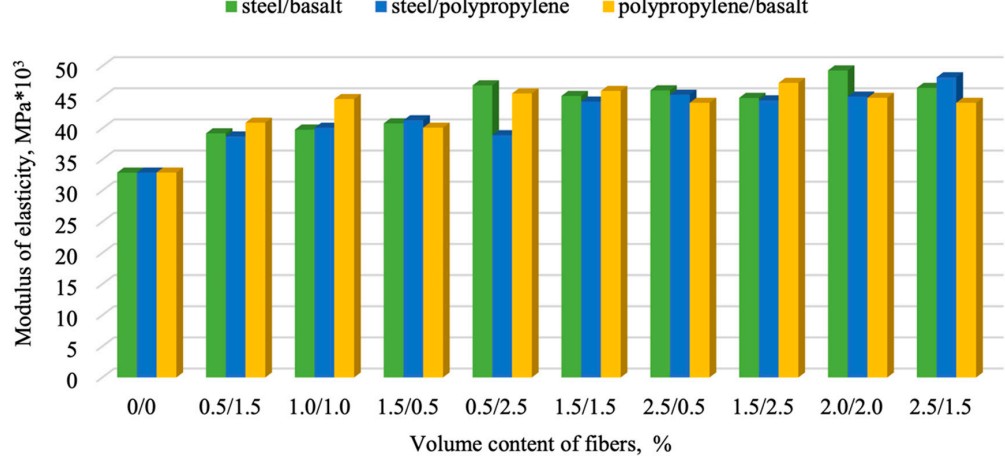

**Figure 11.** Dependence of the modulus of elasticity of fiber-reinforced concrete on the type of combination of fibers and volume content.

Figures 7–11 show that the highest compressive and tensile strengths in bending and elastic modulus, as well as the smallest ultimate deformations under axial compression and tension of fiber-reinforced concrete, are observed with the following combinations and volumetric contents of fiber fibers: steel–basalt (2%–2%), steel–polypropylene (2.5%–1.5%), polypropylene–basalt (1.5%–2.5%).

### 3.2. Regression Equations

Table 9 shows the calculated coefficients of the regression equations for the prototypes of fiber-reinforced concrete.

**Table 9.** Calculated coefficients of regression equations for prototypes of fiber-reinforced concrete.

| The Title of the Output Parameter of the Equation | | Equation Number | Equation Coefficients | | | | | |
|---|---|---|---|---|---|---|---|---|
| | | | $R_0$ | k | $\mu$ | $\mu \cdot k$ | $k^2$ | $\mu^2$ |
| $R_c$ | Compressive strength, MPa | 1 | 52.57 | −0.1241 | 0.2649 | −0.5250 | 2.0421 | −4.0579 |
| | | 2 | 56.92 | −0.2389 | 0.3408 | −1.2000 | −0.9252 | −0.3752 |
| | | 3 | 56.80 | −0.0574 | 0.2167 | 1.4750 | 4.2415 | −0.0585 |
| $R_{tben}$ | Tensile strength in bending test. MPa | 1 | 19.70 | 0.6668 | 0.1908 | 0.4250 | 5.1970 | −13.1530 |
| | | 2 | 17.21 | −0.0908 | 0.1704 | −0.9750 | −0.1859 | −0.6359 |
| | | 3 | 16.96 | 0.0259 | 0.1445 | 0.0500 | 1.5308 | −1.0692 |
| $\varepsilon_{bR}$ | Ultimate deformations during axial compression. mm/m $*10^{-3}$ | 1 | 1.36 | 0.0031 | −0.0109 | −0.0150 | 0.1781 | −0.0519 |
| | | 2 | 1.29 | 0.0035 | −0.0132 | 0.0400 | 0.0881 | 0.0481 |
| | | 3 | 1.29 | −0.0007 | −0.0087 | −0.0075 | −0.0902 | 0.0448 |
| $\varepsilon_{btR}$ | Ultimate deformations during axial tension. mm/m $*10^{-4}$ | 1 | 0.71 | 0.0037 | −0.0087 | −0.0025 | 0.1132 | −0.0118 |
| | | 2 | 0.64 | 0.0033 | −0.0078 | 0.0250 | 0.0232 | 0.0132 |
| | | 3 | 0.65 | −0.0004 | −0.0063 | −0.0025 | −0.0934 | 0.0466 |
| $E_{fc}$ | Elastic modulus. MPa $*10^3$ | 1 | 42.99 | −0.1648 | 0.3315 | 0.0250 | −2.4231 | −1.2231 |
| | | 2 | 44.38 | −0.0889 | 0.2723 | −1.1250 | −1.2400 | −1.1900 |
| | | 3 | 44.07 | 0.0278 | 0.3075 | 0.3000 | 1.9434 | −1.7066 |

Based on the results of studies by the least squares method, the basic regression equations were obtained, which are presented in the form of polynomials of the 2nd degree:

$$R_{c1}(k, \mu) = 52.57 - 0.1241 \cdot k + 0.2649 \cdot \mu - 0.525 \cdot \mu \cdot k + 2.0421 \cdot k^2 - 4.0579 \cdot \mu^2 \quad (3)$$

$$R_{c2}(k, \mu) = 56.92 - 0.2389 \cdot k + 0.3408 \cdot \mu - 1.2 \cdot \mu \cdot k - 0.9252 \cdot k^2 - 0.3752 \cdot \mu^2 \quad (4)$$

$$R_{c3}(k, \mu) = 56,8 - 0.0574 \cdot k + 0.2167 \cdot \mu + 1.475 \cdot \mu \cdot k + 4.2415 \cdot k^2 - 0.0585 \cdot \mu^2 \quad (5)$$

$$R_{tben1}(k, \mu) = 19.7 + 0.6668 \cdot k + 0.1908 \cdot \mu + 0.425 \cdot \mu \cdot k + 5.197 \cdot k^2 - 13.153 \cdot \mu^2 \quad (6)$$

$$R_{tben2}(k, \mu) = 17.21 - 0.0908 \cdot k + 0.1704 \cdot \mu - 0.975 \cdot \mu \cdot k - 0.1859 \cdot k^2 - 0.6359 \cdot \mu^2 \quad (7)$$

$$R_{tben3}(k, \mu) = 16.96 + 0.0259 \cdot k + 0.1445 \cdot \mu + 0.05 \cdot \mu \cdot k + 1.5308 \cdot k^2 - 1.0692 \cdot \mu^2 \quad (8)$$

$$\varepsilon_{bR1}(k, \mu) = 1.36 + 0.0031 \cdot k - 0.0109 \cdot \mu - 0.015 \cdot \mu \cdot k + 0.1781 \cdot k^2 - 0.0519 \cdot \mu^2 \quad (9)$$

$$\varepsilon_{bR2}(k, \mu) = 1.29 + 0.0035 \cdot k - 0.0132 \cdot \mu + 0,04 \cdot \mu \cdot k + 0.0881 \cdot k^2 + 0.0481 \cdot \mu^2 \quad (10)$$

$$\varepsilon_{bR3}(k, \mu) = 1.29 - 0.0007 \cdot k - 0.0087 \cdot \mu - 0.0075 \cdot \mu \cdot k - 0.0902 \cdot k^2 + 0.0448 \cdot \mu^2 \quad (11)$$

$$\varepsilon_{btR1}(k, \mu) = 0.71 + 0.0037 \cdot k - 0.0087 \cdot \mu - 0.0025 \cdot \mu \cdot k + 0.1132 \cdot k^2 - 0.0118 \cdot \mu^2 \quad (12)$$

$$\varepsilon_{btR2}(k, \mu) = 0.64 + 0.0033 \cdot k - 0.0078 \cdot \mu + 0.025 \cdot \mu \cdot k + 0.0232 \cdot k^2 + 0.0132 \cdot \mu^2 \quad (13)$$

$$\varepsilon_{btR3}(k, \mu) = 0.65 - 0.0004 \cdot k - 0.0063 \cdot \mu - 0.0025 \cdot \mu \cdot k - 0.0934 \cdot k^2 + 0.0466 \cdot \mu^2 \quad (14)$$

$$E_{fc1}(k, \mu) = 42.99 - 0.1648 \cdot k + 0.3315 \cdot \mu + 0.025 \cdot \mu \cdot k - 2.4231 \cdot k^2 - 1.2231 \cdot \mu^2 \quad (15)$$

$$E_{fc2}(k, \mu) = 44.38 - 0.0889 \cdot k + 0.2723 \cdot \mu - 1.125 \cdot \mu \cdot k - 1.24 \cdot k^2 - 1.19 \cdot \mu^2 \quad (16)$$

$$E_{fc3}(\mathrm{k}, \mu) = 44.07 + 0.0278 \cdot \mathrm{k} + 0.3075\mu + 0.3 \cdot \mu \cdot \mathrm{k} + 1.9434 \cdot \mathrm{k}^2 - 1.7066 \cdot \mu^2 \quad (17)$$

### 3.3. Graphical Interpretation of Regression Equations

The example of mathematical dependencies is illustrated in Figures 12–14.

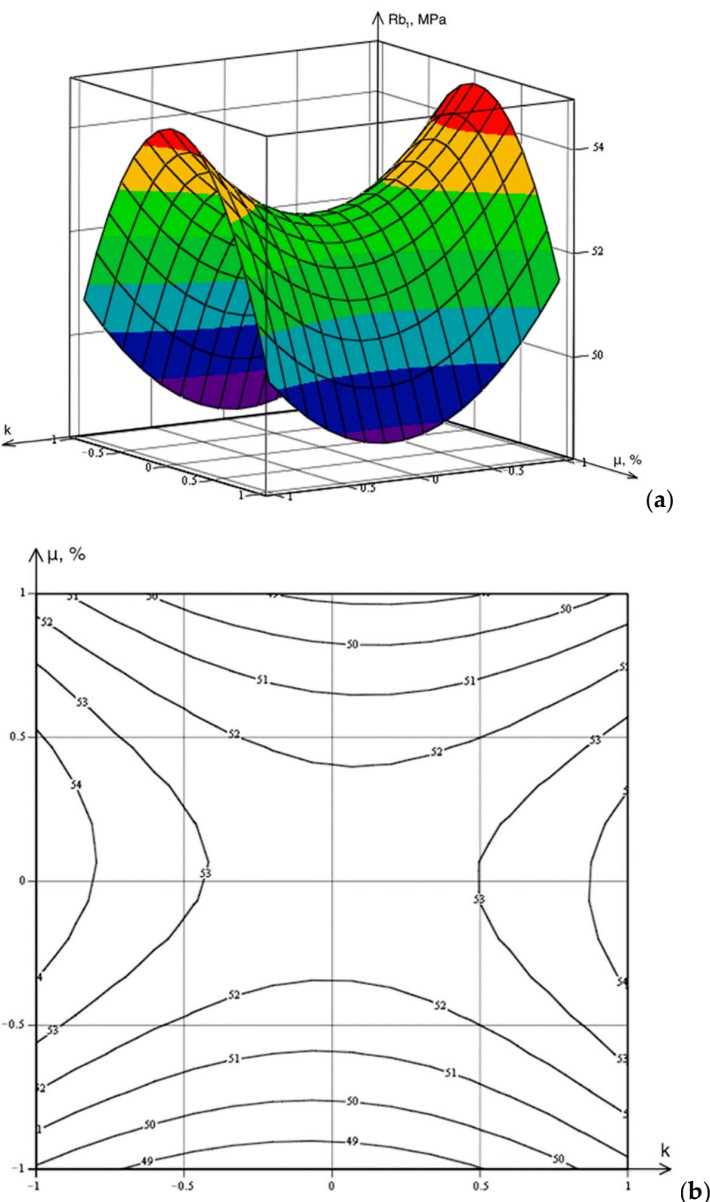

**Figure 12.** Dependence of the compressive strength of concrete reinforced with a combination of steel and basalt fiber (**a**)—three-dimensional surface; (**b**)—isolines.

Based on the results of experimental studies of the effect of a combination of various types of fibers and volume content on the strength and deformation characteristics of fiber-reinforced concrete, the optimal volume concentrations were determined for each type of fiber combination. When using a combination of steel–basalt fiber, the best strength and deformation characteristics were recorded with the volume content of steel fiber 2% and basalt 2%. When using a combination of steel–polypropylene fiber, the best strength and deformation characteristics were recorded with the volume content of steel fiber 2.5% and polypropylene 1.5%. In the combination of polypropylene–basalt fiber, the best strength and deformation characteristics were recorded with the volume content of polypropylene fiber 1.5% and basalt fiber 2.5%.

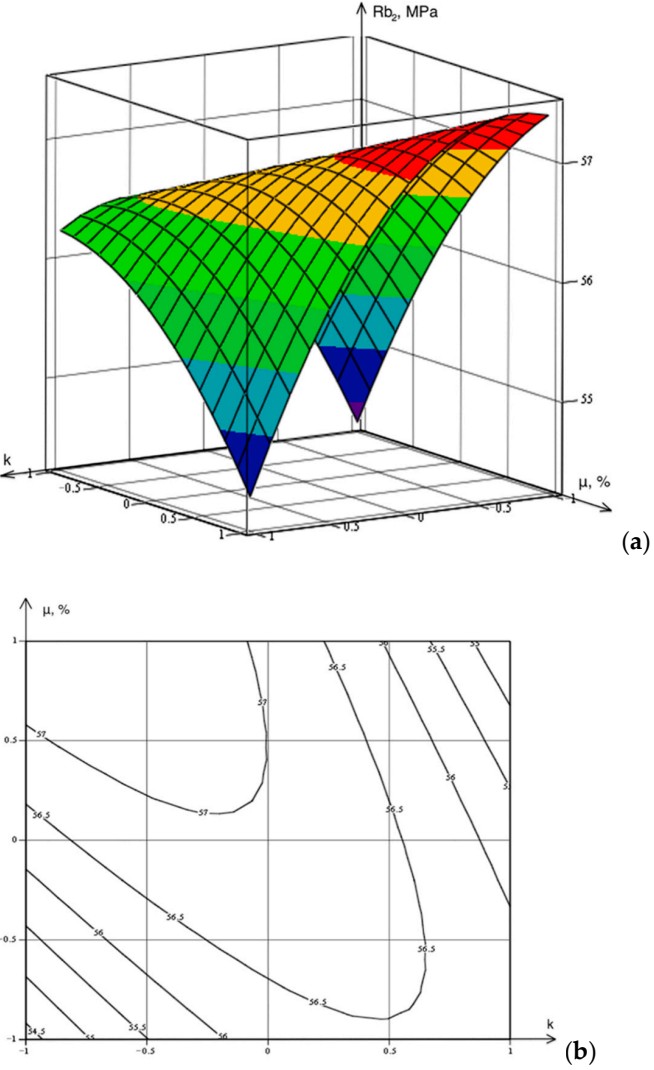

**Figure 13.** Dependence of the compressive strength of concrete reinforced with a combination of steel and polypropylene fiber (**a**)—three-dimensional surface; (**b**)—isolines.

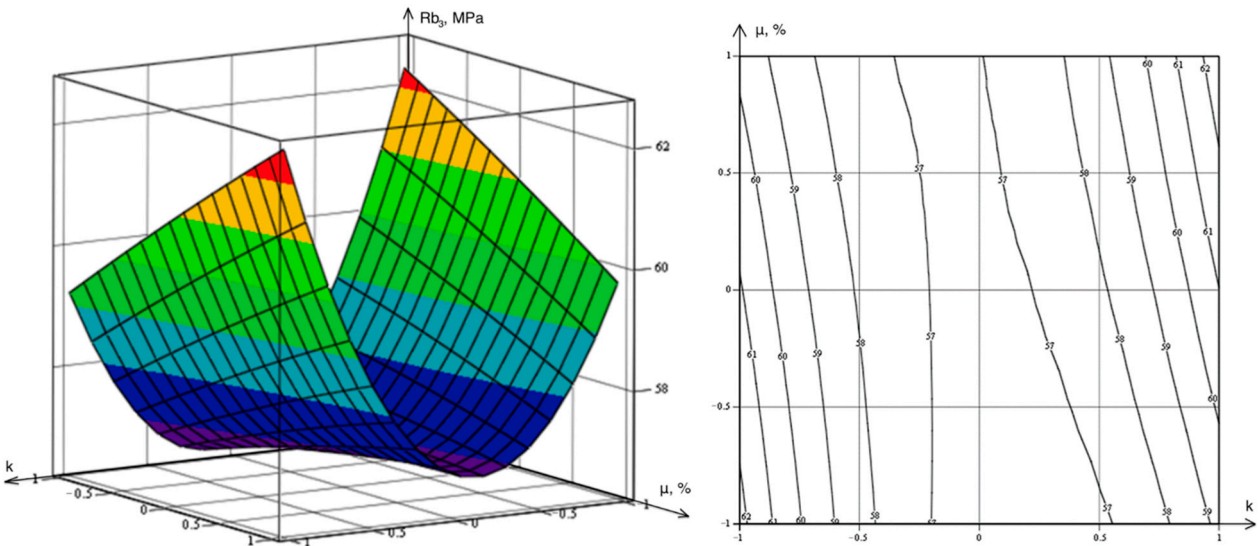

**Figure 14.** Dependence of the compressive strength of concrete reinforced with a combination of basalt and polypropylene fiber.

After analyzing the obtained experimental data and comparing the strength and deformation characteristics of the control samples with the values of the strength and deformative characteristics of the prototypes of fiber-reinforced concrete, the increments of the investigated characteristics were determined. The increase in compressive strength in fiber-reinforced concrete made using a combination of steel–basalt fiber with a volume concentration of steel fiber of 2% and basalt fiber of 2% was 35%, tensile strength in bending increased by 79%, ultimate deformations during axial compression decreased by 52%, ultimate deformations under axial tension decreased by 39%, and elastic modulus increased by 33%. The increase in compressive strength in fiber-reinforced concrete made using a combination of steel–polypropylene fiber with a volume concentration of steel fiber of 2.5% and polypropylene fiber of 1.5% was 34%, tensile strength in bending increased by 78%, ultimate deformations during axial compression decreased by 54%, ultimate deformations under axial tension decreased by 41%, and the modulus of elasticity increased by 31%. The increase in compressive strength in fiber-reinforced concrete made using a combination of polypropylene–basalt fiber with a volume concentration of polypropylene fiber of 1.5% and basalt fiber of 2.5% was 33%, tensile strength in bending increased by 77%, ultimate deformations during axial compression decreased by 56%, ultimate deformations during axial tension decreased by 42%, and the modulus of elasticity increased by 30%.

### 3.4. Deformation Diagrams of Concrete and Fiber-Reinforced Concrete

Moreover, based on the results of experimental studies, calculations were made and diagrams of compression "$\varepsilon_b - \sigma_b$" and tension "$\varepsilon_{bt} - \sigma_{bt}$" were plotted. The following types of concrete state diagrams were used as calculated diagrams of concrete that determine the relationship between stresses and relative deformations: curvilinear, including those with a falling branch, piecewise-linear (two-line and three-line).

Compression and tension diagrams differ significantly in their parameters: concrete tensile strength is an order of magnitude less than compression. To construct diagrams of concrete deformation, we applied the method of constructing curvilinear diagrams [52]. To build a diagram, it is necessary to create an array of points, which is the dependence of the individual stress level, calculated through the entire algorithm for constructing a diagram of concrete deformation, taking into account the ascending or descending branch. Within the framework of the model of short-term loading under axial compression and tension, the concrete deformation diagram is represented as

$$\varepsilon_{b(bt)} = \frac{\sigma_{b(bt)}}{E \cdot v_{b(bt)}} \tag{18}$$

where $\sigma_{b(bt)}$ are stresses arising in concrete under load, $\varepsilon_{b(bt)}$ are ultimate deformations of concrete, $E$ is the modulus of elasticity, and $v_b$ is the coefficient of change of the secant modulus.

The multiplication $E \cdot v_{b(bt)}$ forms the secant module required to construct a curvilinear diagram. For the ascending branch of the diagram, the coefficient of change of the secant modulus is:

$$v_b = \overline{v_{b(bt)}} + (v_0 - \overline{v_{b(bt)}}) * \sqrt{1 - \omega_1 * \eta - \omega_2 * \eta^2} \tag{19}$$

where $\eta = \sigma_{b(bt)} / \overline{b_{(bt)}}$ stress level factor, $v_0 = 1$ secant modulus factor at the beginning of the diagram, and $\varepsilon_{b(bt)} \leq \overline{\varepsilon_{b(bt)}}$, $\omega_1$, $\omega_2$ are variables that determine the curvature of the diagram, depending on the limiting value of the secant modulus at the top of the diagram.

For an ascending curve, the variables $\omega_1$, $\omega_2$ are determined by the formulas

$$\omega_1 = 2 - 2.5 \cdot \overline{\varepsilon_{b(bt)}}; \ \omega_2 = 1 - \omega_1 \tag{20}$$

The principle of constructing the descending branch has a slight difference, the coefficient of change of the secant modulus has the form

$$v_b = \overline{v_{b(bt)}} - (v_0 - \overline{v_{b(bt)}}) * \sqrt{1 - \omega_1 * \eta * - * \eta^2} \tag{21}$$

where $v_0$ is the value of the secant modulus at the beginning of the diagram that differs from the value when constructing the ascending branch and is calculated by the formula

$$v_0 = 2.05 \cdot \overline{v_{b(bt)}} \; ppI \; \varepsilon_{b(bt)} > \overline{\varepsilon_{b(bt)}} \tag{22}$$

The variables $\omega_1$, $\omega_2$, which determine the curvature of the diagram for the descending branch, are determined by the formula

$$\omega_1 = 1.95 \cdot \overline{v_{b(bt)}} - 0.138; \; \omega_2 = 1 - \omega_1 \tag{23}$$

Table 10 shows the calculated coefficients for plotting compression "$\varepsilon_b - \sigma_b$" and tension "$\varepsilon_{bt} - \sigma_{bt}$".

Figure 15 shows diagrams of compression "$\varepsilon_b - \sigma_b$" and tension "$\varepsilon_{bt} - \sigma_{bt}$" for concrete of the control composition and for fiber-reinforced concrete with the best strength and deformation characteristics.

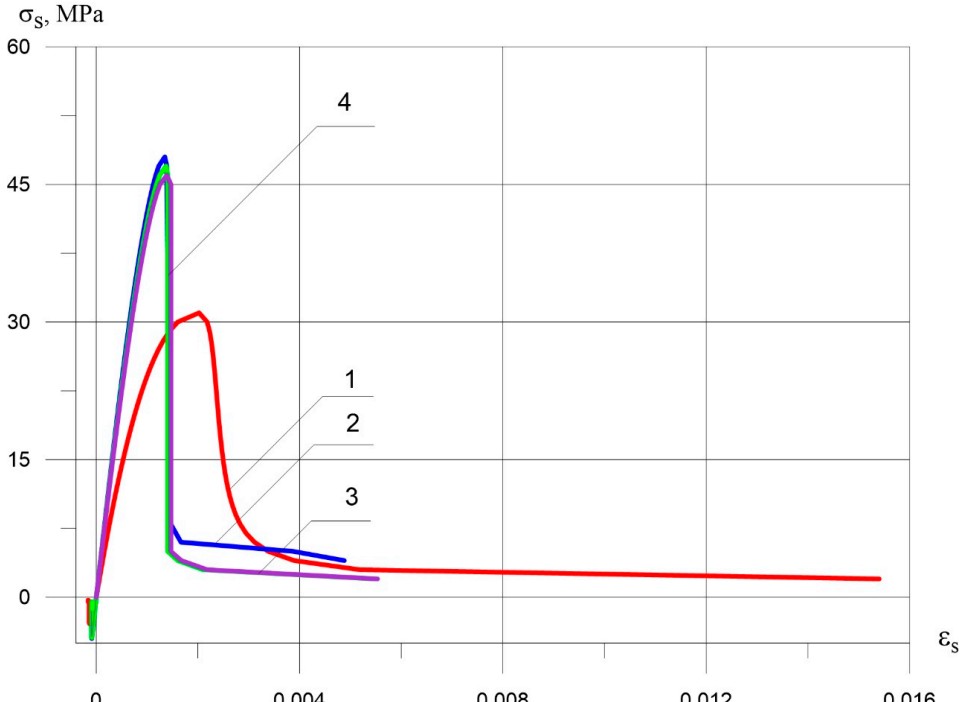

**Figure 15.** Dependencies of the concrete deformation: 1—control composition; 2—concrete with a volume content of steel fiber 2% and basalt 2%; 3—concrete with a volume content of polypropylene fiber 1.5% and basalt 2.5%; 4—concrete with a volume content of steel fiber 2.5% and polypropylene 1.5%.

As a result of the study of the deformations of the control concrete compositions, the following was established: based on the obtained deformation diagrams, the peak of the diagram with displacement up and to the left is present in the deformation diagram of concrete with the volumetric content of steel fiber in the amount of 2% and basalt in the amount of 2%. The rest of the diagrams had a lower peak with a shift to the right. Thus, based on the analysis of the data obtained on the deformation of the tested samples, we conclude that the recommended compositions of fiber-reinforced concrete are advisable.

### 3.5. Recommended Compositions of Various Types of Fiber-Reinforced Concrete

As a result of the experiments, the compositions of heavy and fine-grained concrete were selected using a combination of various types of fiber, with the most effective volume concentrations of various types of fiber among themselves. Table 11 shows the composition options.

**Table 10.** The calculated coefficients for plotting compression "$\varepsilon_b - \sigma_b$" and tension "$\varepsilon_{bt} - \sigma_{bt}$".

| E, *10^4 MPa | $\overline{\sigma}_{b(bt)}$, MPa | $\overline{\varepsilon}_{b(bt)}$, mm/m*10^{-3}/ mm/m*10^{-4} | Ascending Branch | | | | Drop Branch | | | |
|---|---|---|---|---|---|---|---|---|---|---|
| | | | $\overline{v}_{b(bt)}$ | $v_0$ | $\omega_1$ | $\omega_2$ | $\overline{v}_{b(bt)}$ | $v_0$ | $\omega_1$ | $\omega_2$ |
| | | | | | «$\varepsilon_b - \sigma_b$» | | | | | |
| 3.29 | 30.8 | 2.11 | 0.4660 | 1 | 0.8349 | 0.1651 | 0.4660 | 1.7250 | −0.7250 | 0.9554 |
| 3.92 | 37.3 | 1.42 | 0.5635 | 1 | 0.5911 | 0.4089 | 0.5635 | 2.1148 | −1.1148 | 1.1553 |
| 3.98 | 40.7 | 1.51 | 0.6157 | 1 | 0.4607 | 0.5393 | 0.6157 | 2.3233 | −1.3233 | 1.2622 |
| 4.08 | 42.6 | 1.36 | 0.6441 | 1 | 0.3899 | 0.6101 | 0.6441 | 2.4366 | −1.4366 | 1.3203 |
| 3.87 | 36.5 | 1.56 | 0.5522 | 1 | 0.6195 | 0.3805 | 0.5522 | 2.0694 | −1.0694 | 1.1320 |
| 4.01 | 38.6 | 1.48 | 0.5840 | 1 | 0.5401 | 0.4599 | 0.5840 | 2.1964 | −1.1964 | 1.1971 |
| 4.13 | 40.4 | 1.33 | 0.6100 | 1 | 0.4749 | 0.5251 | 0.6100 | 2.3006 | −1.3006 | 1.2506 |
| 4.49 | 43.4 | 1.23 | 0.6565 | 1 | 0.3587 | 0.6413 | 0.6565 | 2.4865 | −1.4865 | 1.3459 |
| 4.93 | 47.8 | 1.11 | 0.7223 | 1 | 0.1943 | 0.8057 | 0.7223 | 2.7494 | −1.7494 | 1.4807 |
| 4.65 | 44.8 | 1.21 | 0.6769 | 1 | 0.3077 | 0.6923 | 0.6769 | 2.5681 | −1.5681 | 1.3877 |
| 4.45 | 41.1 | 1.31 | 0.6214 | 1 | 0.4466 | 0.5534 | 0.6214 | 2.3459 | −1.3459 | 1.2738 |
| 4.51 | 42.1 | 1.24 | 0.6361 | 1 | 0.4097 | 0.5903 | 0.6361 | 2.4049 | −1.4049 | 1.3040 |
| 4.82 | 47.0 | 1.15 | 0.7098 | 1 | 0.2255 | 0.7745 | 0.7098 | 2.6995 | −1.6995 | 1.4551 |
| 4.69 | 41.6 | 1.52 | 0.6293 | 1 | 0.4267 | 0.5733 | 0.6293 | 2.3777 | −1.3777 | 1.2901 |
| 4.52 | 43.7 | 1.25 | 0.6611 | 1 | 0.3473 | 0.6527 | 0.6611 | 2.5046 | −1.5046 | 1.3552 |
| 4.61 | 44.8 | 1.22 | 0.6769 | 1 | 0.3077 | 0.6923 | 0.6769 | 2.5681 | −1.5681 | 1.3877 |
| 3.89 | 39.7 | 1.47 | 0.5998 | 1 | 0.5004 | 0.4996 | 0.5998 | 2.2598 | −1.2598 | 1.2297 |
| 4.43 | 41.9 | 1.34 | 0.6327 | 1 | 0.4182 | 0.5818 | 0.6327 | 2.3913 | −1.3913 | 1.2971 |
| 4.54 | 42.5 | 1.27 | 0.6429 | 1 | 0.3927 | 0.6073 | 0.6429 | 2.4321 | −1.4321 | 1.3180 |
| 4.09 | 36.1 | 1.34 | 0.5454 | 1 | 0.6365 | 0.3635 | 0.5454 | 2.0423 | −1.0423 | 1.1181 |
| 4.47 | 41.6 | 1.29 | 0.6282 | 1 | 0.4296 | 0.5704 | 0.6282 | 2.3731 | −1.3731 | 1.2878 |
| 4.01 | 40.4 | 1.45 | 0.6112 | 1 | 0.4721 | 0.5279 | 0.6112 | 2.3052 | −1.3052 | 1.2529 |
| 4.73 | 46.4 | 1.19 | 0.7007 | 1 | 0.2481 | 0.7519 | 0.7007 | 2.6632 | −1.6632 | 1.4365 |
| 4.49 | 44.9 | 1.22 | 0.6781 | 1 | 0.3048 | 0.6952 | 0.6781 | 2.5726 | −1.5726 | 1.3900 |
| 4.41 | 40.4 | 1.31 | 0.6112 | 1 | 0.4721 | 0.5279 | 0.6112 | 2.3052 | −1.3052 | 1.2529 |
| 4.56 | 43.1 | 1.26 | 0.6509 | 1 | 0.3729 | 0.6271 | 0.6509 | 2.4638 | −1.4638 | 1.3343 |
| 4.6 | 43.7 | 1.19 | 0.6611 | 1 | 0.3473 | 0.6527 | 0.6611 | 2.5046 | −1.5046 | 1.3552 |
| 4.41 | 40.6 | 1.28 | 0.6134 | 1 | 0.4664 | 0.5336 | 0.6134 | 2.3142 | −1.3142 | 1.2575 |
| | | | | | «$\varepsilon_{bt} - \sigma_{bt}$» | | | | | |
| 3.29 | 3.61 | 1.33 | 2.4969 | 1 | 0.3522 | 0.6478 | 0.6591 | 2.4969 | −1.4969 | 1.3512 |
| 3.92 | 3.77 | 0.78 | 3.0482 | 1 | 0.0074 | 0.9926 | 0.7971 | 3.0482 | −2.0482 | 1.6340 |
| 3.98 | 4.05 | 0.73 | 3.3431 | 1 | −0.1771 | 1.1771 | 0.8708 | 3.3431 | −2.3431 | 1.7852 |
| 4.08 | 4.5 | 0.71 | 3.5034 | 1 | −0.2773 | 1.2773 | 0.9109 | 3.5034 | −2.5034 | 1.8674 |
| 3.87 | 4.18 | 0.85 | 2.9841 | 1 | 0.0474 | 0.9526 | 0.7810 | 2.9841 | −1.9841 | 1.6011 |
| 4.01 | 3.84 | 0.72 | 3.1636 | 1 | −0.0648 | 1.0648 | 0.8259 | 3.1636 | −2.1636 | 1.6931 |
| 4.13 | 3.93 | 0.70 | 3.3111 | 1 | −0.1570 | 1.1570 | 0.8628 | 3.3111 | −2.3111 | 1.7688 |
| 4.49 | 4.4 | 0.63 | 3.5739 | 1 | −0.3214 | 1.3214 | 0.9286 | 3.5739 | −2.5739 | 1.9036 |
| 4.93 | 3.89 | 0.52 | 3.9458 | 1 | −0.5540 | 1.5540 | 1.0216 | 3.9458 | −2.9458 | 2.0942 |
| 4.65 | 4.08 | 0.57 | 3.6893 | 1 | −0.3936 | 1.3936 | 0.9574 | 3.6893 | −2.6893 | 1.9627 |
| 4.45 | 4.18 | 0.69 | 3.3752 | 1 | −0.1971 | 1.1971 | 0.8788 | 3.3752 | −2.3752 | 1.8016 |
| 4.51 | 3.70 | 0.61 | 3.4585 | 1 | −0.2492 | 1.2492 | 0.8997 | 3.4585 | −2.4585 | 1.8444 |
| 4.82 | 3.91 | 0.55 | 3.8752 | 1 | −0.5098 | 1.5098 | 1.0039 | 3.8752 | −2.8752 | 2.0581 |
| 4.69 | 3.97 | 0.73 | 3.4201 | 1 | −0.2252 | 1.2252 | 0.8901 | 3.4201 | −2.4201 | 1.8247 |
| 4.52 | 3.37 | 0.60 | 3.5996 | 1 | −0.3374 | 1.3374 | 0.9350 | 3.5996 | −2.5996 | 1.9167 |
| 4.61 | 3.88 | 0.58 | 3.6893 | 1 | −0.3936 | 1.3936 | 0.9574 | 3.6893 | −2.6893 | 1.9627 |
| 3.89 | 3.77 | 0.80 | 3.2534 | 1 | −0.1209 | 1.1209 | 0.8484 | 3.2534 | −2.2534 | 1.7392 |
| 4.43 | 4.3 | 0.70 | 3.4393 | 1 | −0.2372 | 1.2372 | 0.8949 | 3.4393 | −2.4393 | 1.8345 |
| 4.54 | 4.19 | 0.59 | 3.4970 | 1 | −0.2733 | 1.2733 | 0.9093 | 3.4970 | −2.4970 | 1.8641 |
| 4.09 | 3.77 | 0.72 | 2.9457 | 1 | 0.0715 | 0.9285 | 0.7714 | 2.9457 | −1.9457 | 1.5814 |
| 4.47 | 4.02 | 0.69 | 3.4137 | 1 | −0.2212 | 1.2212 | 0.8885 | 3.4137 | −2.4137 | 1.8214 |
| 4.01 | 4.08 | 0.75 | 3.3175 | 1 | −0.1610 | 1.1610 | 0.8644 | 3.3175 | −2.3175 | 1.7721 |
| 4.73 | 3.79 | 1.33 | 3.8240 | 1 | −0.4778 | 1.4778 | 0.9911 | 3.8240 | −2.8240 | 2.0318 |
| 4.49 | 3.61 | 0.78 | 3.6957 | 1 | −0.3976 | 1.3976 | 0.9590 | 3.6957 | −2.6957 | 1.9660 |
| 4.41 | 3.77 | 0.73 | 3.3175 | 1 | −0.1610 | 1.1610 | 0.8644 | 3.3175 | −2.3175 | 1.7721 |
| 4.56 | 4.05 | 0.71 | 3.5419 | 1 | −0.3014 | 1.3014 | 0.9205 | 3.5419 | −2.5419 | 1.8871 |
| 4.6 | 4.5 | 0.85 | 3.5996 | 1 | −0.3374 | 1.3374 | 0.9350 | 3.5996 | −2.5996 | 1.9167 |
| 4.41 | 4.18 | 0.72 | 3.3303 | 1 | −0.1691 | 1.1691 | 0.8676 | 3.3303 | −2.3303 | 1.7786 |

In [53], the results of an experimental study of workability and mechanical properties of self-compacting concrete (SCC) with silica fume (SF) and various types of fibers were obtained. The authors used five types of fibers: hook-end steel (H), mild steel (M), carved steel (S), basalt rock (R), and polypropylene (P) fibers. Each fiber type was added to the concrete at 0.25% by volume of concrete, and a silica fume substitution rate of 30% by

weight of cement was applied. The results showed that the addition of fiber tends to reduce the quality of the fresh concrete. The corresponding compressive strength values were 34.5, 33.9, 32.1, 36.9, 35.8 and 33.3 MPa with the addition of fibers of types H, M, S, P and R at a test age of 28 days, respectively.

**Table 11.** Variants of compositions of various types of fiber-reinforced concrete.

| Concrete Type | Portland Cement, kg/m³ | Sand, kg/m³ | Crushed Stone, kg/m³ | Concrete Components | | | Water, l | Plasticizer, l |
| | | | | Steel Fiber, kg | Basalt Fiber, kg | Polypropylene Fiber, kg | | |
|---|---|---|---|---|---|---|---|---|
| Heavy concrete | 380 | 730 | 1060 | 156 | 56 | - | 195 | 2.28 |
| | 380 | 730 | 1060 | 195 | - | 13.7 | 195 | 2.28 |
| | 380 | 730 | 1060 | - | 72.5 | 13.7 | 195 | 2.28 |
| Fine-grained concrete | 650 | 1450 | - | 156 | 56 | - | 228 | 4 |
| | 650 | 1450 | - | 195 | - | 13.7 | 228 | 4 |
| | 650 | 1450 | - | - | 72.5 | 13.7 | 228 | 4 |

In [54], the strength properties of fine-grained concrete reinforced with amorphous fiber and fiber based on mineral wool, basalt fiber, glass fiber, steel and polypropylene fibers were investigated. The analysis of the results was carried out by comparison with the characteristics of control samples without reinforcement. The best characteristics of the flexural strength were shown by the samples with amorphous fiber, and the highest compressive strength, with the steel fiber. The addition of amorphous fibers increased flexural strength by 56%, but decreased compressive strength by 30% compared to the control samples. The addition of steel fiber shows a 20% increase in flexural strength and a 14% increase in compressive strength, confirming the beneficial effect of adding commercially available fiber to fine concrete. The results obtained allowed the authors to develop compositions of fiber-reinforced concrete with a compressive strength of up to 38 MPa and a bending strength of up to 12 MPa.

A review [55] examined concrete reinforced with steel fiber (SF), polypropylene fiber (PPF), basalt fiber (BF), and glass fiber (GF). Fiber reinforcement in recycled aggregate concrete (RAC) tends to enhance and retard crack propagation and thus lead to plastic behavior of the cementitious matrix. The authors emphasize that despite the growing interest in the use of FRC, there are still some doubts about the dosage and enhancing effects of the fibers.

The authors note that SF is the most primitive and convincing material that shows promising results in terms of increasing the strength of RAC. SF volume fractions up to 0.7% improve compressive strength, while SF volume fractions up to 1.5% result in significant increases in flexural strength and tensile strength RAC. PPF volume fractions in the range of 0–1.25% successfully lead to increased compressive strength, flexural strength at load, and tensile strength RAC. BF is widely recognized as an environmentally friendly twenty-first century fiber that effectively bridges cracks in weak RACs and results in increased tensile strength. BF volume fractions in the 0–1.5% range lead to significant increases in compressive strength, flexural strength at load and tensile strength RAC.

Comparison of the data obtained in [53–55] shows that the results obtained in this article are in good agreement with the results of other researchers.

## 4. Conclusions

Improvement of the technology of heavy concrete is possible in the direction of combined dispersed reinforcement, that is, it is necessary to develop the already known methods of fiber reinforcement of heavy concrete in the direction of combined reinforcement, in order to significantly increase the structural characteristics of the resulting concretes. As a result of the work detailed in this article, large-scale studies were carried out, as a result of which the most effective combinations of fibers of different types of fibers were determined with each other and their optimal volume concentrations. A comparative assessment is given according to the results of strength and deformation characteristics between samples

of the control composition and samples reinforced with various combinations of fibers, graphs and diagrams of concrete deformation are built, calculations are made, and rational compositions are selected for heavy and fine-grained concrete.

During the work detailed in this article, the following was done:

- Large-scale studies were carried out, as a result of which the most effective combinations of fibers of various types of fibers were determined among themselves and their optimal volumetric concentrations (steel–basalt: 2%–2%; steel–polypropylene: 2.5%–1.5%; polypropylene–basalt: 1.5%–2.5%).
- A comparative assessment is given according to the results of strength and deformation characteristics between samples of the control composition and samples reinforced with various combinations of fibers;
- Built graphs, diagrams of concrete deformation;
- Calculations were made and rational compositions were selected for heavy and fine-grained concrete.

The results of the study are recommended for use in industrial and civil construction in the design and calculation of building structures made of reinforced concrete products of annular cross-section with a variatropic structure, as well as using fiber dispersion-reinforcing fibers and in industrial technologies for the production of such products, in regulatory and technical documents for the calculation, design and manufacture of such structures, for the conditions of a construction site and for use in educational and methodological documents in the study of disciplines on the technology of reinforced concrete products and technology of construction production.

As for the directions for further research on the topic of the combined dispersed reinforcement of heavy concretes, such areas as the nanomodification of heavy concrete, that is, the use of nanomodifying additives, primarily the use of nanosilica, can also be considered, but it is also possible to consider the use of other nanomodifying additives, which will be done in the future. Moreover, our further task is to determine rational ways of introducing fiber fibers into the concrete mixture, namely, to consider the possibility of the stage-by-stage mixing of fiber fibers with the components of the concrete mixture.

**Author Contributions:** Conceptualization, L.R.M., S.A.S., E.M.S., A.N.B. and D.E.; methodology, S.A.S., E.M.S., A.V.S.; software, S.A.S., E.M.S., A.N.B. and D.E.; validation, L.R.M., S.A.S., E.M.S. and A.N.B.; formal analysis, S.A.S., E.M.S. and A.S.S.; investigation, L.R.M., S.A.S., E.M.S., A.N.B., A.V.S. and B.M.; resources, B.M.; data curation, S.A.S., E.M.S.; writing—original draft preparation, S.A.S., E.M.S. and A.N.B.; writing—review and editing, S.A.S., E.M.S. and A.N.B.; visualization, S.A.S., E.M.S., A.N.B. and D.E.; supervision, L.R.M., B.M.; project administration, L.R.M., B.M.; funding acquisition, A.N.B., B.M. All authors have read and agreed to the published version of the manuscript.

**Funding:** This research received no external funding.

**Institutional Review Board Statement:** Not applicable.

**Informed Consent Statement:** Not applicable.

**Data Availability Statement:** The study did not report any data.

**Conflicts of Interest:** The authors declare no conflict of interest.

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
