# Peer review of "Improving the Structural Characteristics of Heavy Concrete by Combined Disperse Reinforcement"

_applsci, doi:10.3390/app11136031_

Round 1
Reviewer 1 Report
Dear authors,
Thank you very much for this important article that reflects the needs of Heavy Concrete and its improvements.
The title and abstract are fine.
The hypothesis of a combination of different fibers is very interesting.
Literature and introduction are very well prepared.
I might recommend using subheadings for clarity because the text is long.
I recommend expanding the literature with an article from the field, for example:
10.1016 / j.jobe.2021.102464
10.3390 / crystals 10030220
10.1016 / j.struct.2020.111031
or find more.
The description of the samples and the test is well done.
But I lack more information about concrete - volume or weight composition would be beneficial.
References to standards are missing for tests and methods - the name is not enough.
The verbal evaluation of the results from pages 9 to 16 is missing, it would be better to add directly to the specific pictures. The article seems inappropriate in this part.
The conclusions should contain nominal results - specification of the evaluation of the obtained data.
A few notes:
- be careful, table 9 should be on one sheet, table 10 as well,
- split Table 11 in two,
- you have nice graphs 12 to 15, but graphs 7 to 11 are very unscientific.
- in front of pictures and after pictures (and tables) it is necessary to make a free line.
Regards,
Author Response
Dear Reviewer!
Thanks for your valuable comment!
I'm sure this will make the article better and more readable.
- I might recommend using subheadings for clarity because the text is long.
Thanks for the comments. Subsections 3.1, 3.2, 3.3, 3.4, and 3.5 have been added to the text of the article.
- I recommend expanding the literature with an article from the field, for example: 10.1016/j.jobe.2021.102464; 10.3390/crystals10030220; 10.1016/j.struct.2020.111031 or find more
The links suggested by you and the second reviewer have been added. Corrections in the text are highlighted in red. The link 10.1016/j.struct.2020.111031 could not be found, such DOI does not exist, another one was replaced.
- The description of the samples and the test is well done. But I lack more information about concrete - volume or weight composition would be beneficial
Dear Reviewer, thanks for the comment. The study of the influence of the combination of different types of fibers on the characteristics of fiber-reinforced concrete was carried out on a composition with a component ratio indicated in lines 243-245
All samples were made of fine-grained concrete of the same composition Cement/Sand = 1: 2 at Water/Cement = 0.32 and the consumption of the superplasticizer additive in the amount of 0.6% of the cement mass
- References to standards are missing for tests and methods - the name is not enough
Dear reviewer, thanks for the comment. The names of the test methods for samples are specified, added to the designation of GOST. Links URL added to the list of references.
- The verbal evaluation of the results from pages 9 to 16 is missing, it would be better to add directly to the specific pictures. The article seems inappropriate in this part.
Dear reviewer, thanks for the comment. Added evaluation of results after graphic material (lines 344-348, 484-490)
- The conclusions should contain nominal results - specification of the evaluation of the obtained data.
The Conclusion has been completely revised considering the comments of both reviewers. The nominal results have been added to the Conclusion.
- A few notes: 1 - be careful, table 9 should be on one sheet, table 10 as well; 2 - split Table 11 in two; 3 - you have nice graphs 12 to 15, but graphs 7 to 11 are very unscientific; 4 - in front of pictures and after pictures (and tables) it is necessary to make a free line.
The authors have made the appropriate corrections:
- spacing was added before figures and after figures (and tables).
Tables 9 is on one sheet. The layout of Tables 10 and 11 will be done during editorial revision.
The authors would like to retain Figures 7 - 11 since they clearly show the effect of each additive on the properties of concrete
Finally, we would like to thank you for your valuable comments, attention to our work, and your support. We understand that your comments have made our article better and more readable.

Reviewer 2 Report
The article discuss the topic of the Improving the Structural Characteristics of Heavy Concrete by Combined Disperse Reinforcement. This is an interesting paper that deals with a timely topic and novel idea.
However, in my opinion article should be improved before potential publication. The following modification should be considered:
1. It is worth to add graphical abstract.
2. I suggest to add separated point - Research significance - Please describe here the main essence of the research. What was the inspiration for such an analysis? Why presented studies are so important?
A part of text in lines 175-178 could be useful.
3. It is recommended to study some literature involved with considered scientific area where authors define the major impact of the aggregate shape and content of fibres on the mechanical properties, pore distribution and behavior of fibre reinfored concrete; such as: 10.1007/s10853-016-9917-4 and https://doi.org/10.3390/ma11081372
4. Table 1: Specific surface area (according to Blaine) - please change the unit to m2/kg or m2/g
5. Table 5: How was measured the properties of fibres? If this data were received from manufacturer, please add this information in the text.
6. Please add the scale in case of figures 2,3 and 4.
7. In my opinion Technical characteristics of the concrete mixer BL-10 is redundant.
8. Line 262: what 'Factorial design' means?
9. Table 9: Please change the unit of Elastic modulus to GPa.
10. How were measured ultimate deformations of the specimens? Please explain it in the text.
11. Table 12: Variants of compositions of various types of fiber-reinforced concrete should be presented in part Materials and methods.
12. The conclusion can be more concise. I suggest that conclusions should be presented point by point.
13. It is recommended to indicate potential application of research results in civil engineering or another discipline.
Author Response
Dear Reviewer!
Thanks for your valuable comment!
I'm sure this will make the article better and more readable.
- It is worth to add graphical abstract.
Dear reviewer, thanks for the comment. Photos and Figures from the text of the article will be used as Graphical Abstract.
- I suggest to add separated point - Research significance - Please describe here the main essence of the research. What was the inspiration for such an analysis? Why presented studies are so important? A part of text in lines 175-178 could be useful.
Thanks for the comment. This information is presented in the Introduction. Lines 38-47, 192-199
- It is recommended to study some literature involved with considered scientific area where authors define the major impact of the aggregate shape and content of fibres on the mechanical properties, pore distribution and behavior of fibre reinforced concrete; such as: 10.1007/s10853-016-9917-4 and https://doi.org/10.3390/ma11081372
The links suggested by you and the second reviewer have been added. Corrections in the text are highlighted in red.
- Table 1: Specific surface area (according to Blaine) - please change the unit to m2/kg or m2/g
Corrected. The unit of measurement for the specific surface area was changed to m2/kg.
- Table 5: How was measured the properties of fibres? If this data were received from manufacturer, please add this information in the text
Thanks for the comment. Fiber property data was obtained from the manufacturer. Changes have been made to the text of the article (table 5).
- Please add the scale in case of figures 2,3 and 4.
Thank you for your comment. The scale in Figures 2, 3 and 4 has been added (in figure captions).
- In my opinion Technical characteristics of the concrete mixer BL-10 is redundant.
Agree. Table 7 was removed from the text of the article.
- Line 262: what 'Factorial design' means?
Sorry for the mistake. These are Values varying factors. We made corrections to the text of the article (the title of table 7 - line 314)
- Table 9: Please change the unit of Elastic modulus to GPa
In table 9, the unit of elasticity modulus was changed to GPa
- How were measured ultimate deformations of the specimens? Please explain it in the text
A method for measuring the ultimate deformations of samples was added to the text of the article. Lines 274-289
- Table 12: Variants of compositions of various types of fiber-reinforced concrete should be presented in part Materials and methods
Thank you for your comment. These compositions of various types of fiber-reinforced concrete are the result of work following the development of the recommendations received.
- The conclusion can be more concise. I suggest that conclusions should be presented point by point
Thank you for your comment. The Conclusion section has been redone in accordance with both Reviewers remarks and presented point by point
- It is recommended to indicate potential application of research results in civil engineering or another discipline
Thank you for your comment. Research results in civil engineering and related disciplines has been added to the text of the article as the possible application. Lines 558-565
Finally, we would like to thank you for your valuable comments, attention to our work, and your support. We understand that your comments have made our article better and more readable.

Round 2
Reviewer 2 Report
The majority of remarks have been considered by authors.
The authors responded to comments of the reviewer thoroughly.
Considering the above, I suggest that article could be published.